# Use of New Ultrasonography Methods for Detecting Neoplasms in Dogs and Cats: A Review

**DOI:** 10.3390/ani14020312

**Published:** 2024-01-19

**Authors:** Anna Carolina Mazeto Ercolin, Alex Silveira Uchôa, Luiz Paulo Nogueira Aires, Diego Rodrigues Gomes, Stefany Tagliatela Tinto, Giovanna Serpa Maciel Feliciano, Marcus Antônio Rossi Feliciano

**Affiliations:** 1Laboratory of Veterinary, Faculty of Animal Science and Food Engineering, Sao Paulo University, Sao Paulo 13635-900, Brazil; anna.ercolin@usp.br (A.C.M.E.); uchoaasu@gmail.com (A.S.U.); rodrigues.gomes@usp.br (D.R.G.); stefanytinto@yahoo.com.br (S.T.T.); giovanna_serpa@hotmail.com (G.S.M.F.); 2Department of Veterinary Clinic and Surgery, School of Agricultural and Veterinarian Sciences, Sao Paulo State University “Júlio de Mesquita Filho”, Sao Paulo 14884-900, Brazil; luiz_paulo.aires@hotmail.com

**Keywords:** elastography, contrast-enhanced ultrasound, neoplasia, metastasis, feline, canine

## Abstract

**Simple Summary:**

The aim of this literature review was to present the novel imaging modalities elastography and contrast-enhanced ultrasonography. We provided an overview of the concepts and applications of each technique for the investigation of neoplastic or metastatic tumors in dogs and cats. Studies fon elastography are based on the elasticity and deformation of the evaluated tissue. The information obtained from elastography can aid in the detection and differentiation of malignant and benign structures. Descriptions of elastography studies in several organs and tissues in veterinary medicine reported that, in general, malignant tumors tend to be more rigid and, therefore, less deformable than benign lesions or in comparison to the healthy parenchyma. Contrast-enhanced ultrasonography is based on the intravenous injection of contrast media constituted by microbubbles. This imaging modality provides information on the tissue perfusion and allows the investigation of macro- and micro-circulation. Studies on different organs and tissues were performed in dogs and cats and revealed a tendency of malignant tumors to present faster transit of the contrast media (time to wash in, peak and wash out). These advanced techniques associated with other imaging modalities can be used as screening tests and can potentially represent an alternative to the invasive sampling methods required for cytological and histopathological analysis.

**Abstract:**

The aim of this literature review was to present the novel imaging modalities elastography and contrast-enhanced ultrasonography. We provided an overview of the concepts and applications of each technique for the investigation of neoplastic and metastatic tumors in dogs and cats. Studies on elastography are based on the elasticity and deformation of the evaluated tissue. The information obtained from the different types of elastography can aid in the detection and differentiation of malignant and benign structures. Descriptions of elastography studies in several organs and tissue in veterinary medicine reported that, in general, malignant tumors tend to be more rigid and, therefore, less deformable than benign lesions or in comparison to the healthy parenchyma. Contrast-enhanced ultrasonography is based on the intravenous injection of contrast media constituted by microbubbles. This imaging modality can be performed in nonsedated animals and provides information on the tissue perfusion, allowing the investigation of macro- and micro-circulation. Studies on different organs and tissues were performed in dogs and cats and revealed a tendency of malignant tumors to present faster transit of the contrast media (time to wash-in, peak and wash-out). These advanced techniques can be associated with other imaging modalities, aiding important information to the well-established exams of B-mode and Doppler ultrasonography. They can be used as screening tests, potentially representing an alternative to the invasive sampling methods required for cytological and histopathological analysis.

## 1. Introduction to Elastography

Elastography is a relatively new ultrasonographic technology, created in the 1990s [1], that is noninvasive and used to measure the stiffness or elasticity of tissues [2]. There are two main forms of elastography used as diagnostic methods, static elastography, and dynamic elastography [2]. The authors described static elastography or the strain-wave modality as involving manual pressure from the transducer on the area under study, compressing the tissue to measure the relative tissue displacement. In this technique, the echoes are acquired before and after slight compression of the tissue using the sonographic probe. After compression, a sample of linear echoes are obtained. The data are, therefore, compared with cross-correlation techniques [3]. For this modality, qualitative and semi-quantitative assessments can be performed to estimate the elasticity index by evaluating a color graph (elastogram).

Dynamic elastography encompasses shear-wave elastography techniques. Sonoelastography (Fibroscan^®^, EchoSens, Paris, France) involves low-frequency vibration (20 Hz to 1000 Hz) externally applied to produce internal vibrations in the tissue under study. It allows quantifying the propagation and speed of the shear waves (expressed in kilopascals—kPa). This technique consists of exciting a tissue using harmonic vibration, producing low-frequency and high-amplitude waves. The low frequencies reduce the attenuation, allow deeper penetration, and avoid damages to the neighboring tissues [3]. Shear-wave speed is directly related to tissue stiffness. Therefore, the stiffer the tissue, the faster the shear-wave propagation (higher kPa) [4].

Shear-wave elastography evaluates tissue displacement from a force caused by a focused, high-intensity sound beam that produces shear waves [2]. These waves laterally pass through the tissue at a speed between 1 to 10 m/s and are rapidly attenuated by organic tissues. This method has lower inter-observer variability compared to manual compression elastography [1] and can be performed using commercially available equipment other than Fibroscan^®^, such as Acoustic Radiation Force Impulse (ARFI) (Siemens, Munich, Germany) or SuperSonic Shear-wave Imaging (SuperSonic Imaging, Aix-en-Provence, France) [5]. These techniques permit both qualitative evaluations with an elastogram, representing the degree of tissue elasticity or stiffness in a color graph, and quantitative evaluations, measuring the shear-wave speed. Like sonoelastography, tissue elasticity correlates with the speed of sound-wave propagation, measured in m/s (meters per second) or kPa (kilopascal), where tissues with higher stiffness present higher shear-wave propagation speeds [2]. Shear-wave elastography generates greater penetration of the mechanical waves so that obesity and ascites create no limitations for this exam [6].

Several studies have used elastography for assessing hepatic fibrosis in humans as a preliminary assessment before tissue biopsies or for predicting and detecting malignancy [5,7], as well as for assessing acute and chronic kidney diseases [8]. In veterinary medicine, recent studies have explored elastography for evaluating the prostate [9,10,11], liver [12], and kidneys in dogs and cats [2,9], as well as for assessing hepatic fibrosis [13], lymph nodes [2,14,15,16,17,18,19,20,21,22], and mammary neoplasms [1,4,23,24,25]. Elastography has been shown to be a promising diagnostic method in the evaluation and prediction of neoplasm malignancy, mainly due to its safety and low invasiveness [1].

## 2. Applicability of Elastography

### 2.1. Mammary Glands

Mammary neoplasia is highly prevalent in dogs and can cause severe consequences to the animals [26]. For this reason, its diagnosis and treatment must be very aggressive and effective. Mammary neoplasia presents several molecular and clinicopathological similarities with mammary tumors in women [27]. For that reason, several studies have attempted to differentiate benign and malignant mammary tumors in dogs [1,4].

A study of elastography for canine mammary tumors was able to differentiate benign nodules (such as mammary hyperplasia, adenoma, fibroadenoma and mixed benign tumors) and malignant tumors (such as tubular carcinoma, complex tubular papilliferous carcinoma, mixed carcinoma, simple solid carcinoma, and complex carcinoma) [1]. This study reported that malignant mammary nodules were more rigid and, therefore, presented a higher mean shear-wave velocity (3.33 m/s) when compared to benign nodules (1.28 m/s).

ARFI elastography and other imaging modalities were used to investigate 300 mammary masses in dogs [23]. The authors reported that a shear-wave velocity higher than 2.57 m/s presented a sensitivity of 94.7%, a specificity of 97.2%, and high accuracy for the detection of malignancy. The researchers explained that malignant mammary tumors presented high stiffness and high shear-wave velocity. They said that this event can be explained by the stromal reaction induced by the carcinoma and is associated with increased collagen fibers in the mammary tissue. Similarly, a study on shear-wave elastography reported lower shear-wave velocity values for benign mammary nodules when compared to mammary tumors in dogs [24]. However, these authors highlighted that some benign and mixed malignant tumors can present ossified (more rigid) or cartilaginous tissues (less rigid), which could lead to a misdiagnosis.

Elastography studies involving domestic felines are scarce. Research on ARFI elastography of the mammary tissue of two female cats reported high shear-wave velocity (cat 1 = 4.07 m/s and cat 2 = 4.54 m/s and 6.58 m/s). The qualitative evaluation revealed rigid and nondeformable tissues. Those findings suggested the presence of malignant mammary tumors, confirmed by histopathological analysis as tubular carcinoma and cribriform mammary carcinoma [25].

### 2.2. Lymph Nodes

Lymph node evaluation is paramount to staging oncological patients since the presence of metastatic lymph nodes indicates negative prognostics. The sentinel lymph node is the first draining lymph node, and its location can optimize the success of tissue sampling for histopathology or cytology [28]. Sentinel lymph node mapping has been described for a wide variety of neoplasms, such as mast cell tumors [29,30,31,32,33] and melanoma [29] in dogs and endometrial tumors in women [34].

The diagnosis of lymphadenopathies is routinely performed with fine needle aspiration cytology. Some limitations of this method include the (frequent) insufficient amount of tumoral cells obtained in the samples and the high possibility of sample contamination, producing false negative results. B-mode ultrasonography is the first-choice imaging modality used to screen for lymph node metastasis. However, as with cytology, the ultrasound has some limitations due to the overlap of findings between benign inflammatory and neoplastic etiologies [14].

ARFI elastography was reported to be more sensitive and specific than the short/long axis ratio (evaluated with B-mode ultrasound) for the detection of axillary and inguinal metastatic lymph nodes in bitches with mammary neoplasia. The metastatic lymph nodes of bitches with mammary tumors were more rigid than reactive or normal lymph nodes [15], with an accuracy higher than 95% for the detection of malignant lymphoid tissues.

A recent study utilized ARFI elastography to evaluate dogs with tumors in the head or cervical region. It was observed that mandibular or medial retropharyngeal sentinel lymph nodes showed higher shear-wave propagation velocities, indicating greater tissue stiffness. These values were statistically different in cases of metastasis compared to unaffected sentinel lymph nodes [14]. However, the authors reported low diagnostic sensitivity, supporting studies on head and neck neoplasms in humans [16,17], in which the diagnostic sensitivity of ARFI elastography varied with an increase in the number of included lymph nodes in the assessments. Additionally, focal areas with higher stiffness were observed within the lymph nodes compared to the total area of the lymph node itself, showing that in the early stages of the metastatic process, the lymph node is not diffusely affected, which can be a limiting factor and a source of false-negative results in metastasis research [14].

Another study revealed that benign lymph nodes exhibited lower stiffness than malignant ones. The lymph nodes were classified according to stiffness scores to differentiate between malignant and benign ones, although there was an overlap in the classification between these two groups [18]. The assessment of stiffness scores, performed semi-quantitatively with an elastogram, showed high sensitivity and specificity for detecting malignancies in the mandibular lymph nodes of dogs with head and neck tumors [19].

Strain-wave elastography contributed to the differentiation of metastatic mandibular lymph nodes. The authors suggested an elastography pattern based on the percentage of blue areas (stiff areas), where Grade 1 showed no blue areas, while in Grade 4, the entire lymph node appeared blue. Metastatic mandibular lymph nodes exhibited high-grade elastography patterns [20]. The same authors suggested the association between elastography and contrast-enhanced ultrasonography for the detection of metastatic mandibular lymph nodes. The associations between different imaging modalities were also described in another study, where the parameters with the best accuracy for detecting malignancies in mandibular lymph nodes were the resistivity index (Doppler ultrasound), short-axis size (B-mode ultrasound), and elasticity (elastography). Malignant lymph nodes showed larger dimensions when compared to benign ones, a mixed vascular distribution, and higher resistivity, pulsatility, and elasticity scores than benign nodes [21].

The use of multimodal imaging has been suggested by other authors for the differentiation of malignant mesenteric (lymphoma), fibrotic (eosinophilic sclerosing lymphadenitis), and reactive (reactive nodular hyperplasia) lymph nodes in cats. It was possible to differentiate reactive lymph nodes, which obtained lower scores in an ultrasound classification system and lower stiffness in an elastography examination. However, lymph nodes with lymphoma and lymphadenitis exhibited some degree of overlap in the classification. Some fibrotic lymph nodes scored higher in the ultrasound classification and showed greater stiffness in elastography, like lymph nodes affected by lymphoma [22].

### 2.3. Spleen

Splenic tumors are quite common in small animal clinics, especially in dogs. Approximately 58% of tumors larger than 1 cm in their largest axis are considered malignant, with hemangiosarcoma being the most frequent [35]. However, size, shape, and other characteristics from the B-mode ultrasound examination do not safely allow the differentiation between malignant and benign tumors, as benign and malignant neoplasms often share very similar echotexture and echogenicity patterns [36]. In these cases, elastography can contribute to identifying neoplastic lesions and overcome the limitations of the B-mode, serving as a complementary tool in the study of oncology patients by assessing tissue rigidity.

Strain elastography was used to differentiate malignant and benign hypoechoic splenic lesions smaller than 4 cm in width based on the elasticity index and stiffness value. Malignant lesions presented elasticity rates equal to or greater than 1.5 and stiffness values higher than 70% [37]. These authors explained that the stiffness value is calculated as the percentage of the lesion that is encoded as rigid, whereas the elasticity index can be calculated by the ratio between an area of normal parenchyma and the corresponding area of the entire lesion, considering the assessment of areas of the same size and depth.

A study evaluating 37 spleens of patients with splenic nodules observed that malignant nodules exhibited higher shear-wave velocity compared to benign ones, indicating that malignant lesions are stiffer [36]. This characteristic demonstrated that shear-wave elastography was 97% accurate in the detection of malignant splenic nodules (considered superior to advanced imaging methods such as magnetic resonance imaging). The authors reported that a shear-wave velocity greater than 2.6 m/s was indicative of malignancy in splenic lesions with 95% sensitivity, 100% specificity, and an accuracy of 97%. Examples of elastography images of splenic nodules can be seen in Figure 1 and Figure 2.

### 2.4. Cutaneous Nodules

The cutaneous tissues, like other tissues and organs previously described, benefit from the association between different imaging modalities, such as B-mode, power Doppler and elastography, that contribute to increasing the specificity of the evaluations [38].

Strain elastography was able to differentiate some cutaneous nodules, such as mastocytoma and benign follicular tumors. They presented the highest elasticity scores among the neoplastic nodules. Calcified and nonvascularized nodules presented higher elasticity scores, and there was a negative correlation between the longitudinal diameter of the cutaneous nodules and qualitative elastographic parameters [39].

ARFI elastography associated malignant cutaneous and subcutaneous lesions with nondeformable tissues and shear-wave velocities > 3.52 m/s [38]. Similarly, another study compared lipomas and malignant cutaneous tumors and attributed the higher stiffness score to malignant lesions [40].

### 2.5. Liver

B-mode ultrasonography is the first-choice method for hepatic evaluation in dogs and cats due to its advantages of being a noninvasive, quick and low-cost technique with high sensitivity for the detection of nodular or cystic lesions [12]. The same authors pointed out that the correlation of ultrasonographic findings with laboratorial exams (such as cytology or histopathology) is mandatory to determine the relevance of the imaging findings or to confirm the diagnosis of tumoral lesions and determine the tumoral cell type. In this way, elastography can provide important information, aiding in the differentiation of tumoral tissue and normal hepatic parenchyma.

Hepatic lesions were submitted to a qualitative evaluation using an elastogram, where regions in blue represented rigid tissues, green spots were intermediate, and regions in red corresponded to soft tissues. Malignant hepatic lesions were presented in blue, indicating rigid tissues. In addition, the average intensity of colors in the elastogram was higher in cases of malignant tumors [41].

Elastography can present some limitations for hepatic evaluation, as higher frequency transducers are not able to promote adequate tissue deformation in deeper hepatic regions. Evaluations of deeper lesions or those in deep-chested or large-breed dogs are limited [41].

### 2.6. Prostate and Testes

Ultrasound in its various modalities (such as B-mode and Doppler, for example) has limitations in differentiating prostatic and testicular lesions by producing nonspecific information about these lesions [9]. Therefore, elastography appears to be a promising method, providing additional information for distinguishing different types of lesions in the prostate and testicles of dogs and cats or allowing the delineation of the lesion area for puncture and material collection.

A study of dogs assessed their tissue homogeneity, deformation capacity, and shear-wave velocity, and it revealed that the shear-wave velocity was significantly higher in prostatic alterations when compared to normal prostatic tissue [10]. The same authors suggested that a velocity greater than 2.35 m/s is potentially associated with malignant lesions.

Studies involving elastographic evaluations of prostatic and testicular conditions in domestic cats are scarce. Descriptions of normal elastographic parameters of the prostate and testicles of cats can also contribute to the differentiation of benign and malignant tumors, as malignant tumors are typically characterized as rigid and with high shear-wave velocities [42].

A study involving healthy cats enabled the detection of elastographic particularities in this species when compared to dogs [42]. The testicles of the studied animals exhibited higher shear-wave velocities than those of dogs. This fact can probably be justified by the greater amount of fibrous tissue in the canine testicle.

In a pioneering study involving the evaluation of 18 dogs with testicular diseases using ARFI elastography, neoplastic, inflammatory, and degenerative lesions of the testicular parenchyma were classified as nondeformable and heterogeneous, while normal testicles were homogeneous and nondeformable in a qualitative assessment [43]. In a quantitative assessment, the authors demonstrated that neoplastic testicles had shear-wave velocities of 3.32 ± 0.65 m/s, 2.99 ± 0.07, and 2.73 ± 0.37 for interstitial cell tumors, sertolioma, and leydigoma, respectively. In another study from the same authors [11], when testicles of healthy dogs were evaluated, the reference values found for shear-wave velocity were 1.23 m/s in elderly and young adult patients and 1.28 m/s in juvenile patients (under 1 year of age).

## 3. Introduction to Contrast-Enhanced Ultrasonography (CEUS)

Contrast-enhanced ultrasonography (CEUS) was introduced with more confidence and security in medicine in the 90s to evaluate cardiac perfusion [44]. Currently, this imaging modality is being widely used in medicine and veterinary medicine to evaluate renal tissue perfusion and hepatic, reproductive, and neoplastic tissues, for example [44,45,46,47,48,49].

This imaging modality is based on the use of an intravenous injection of contrast media composed of microbubbles [50]. The same authors stated that more recent products available commercially act exclusively in the intravascular region and are composed of a lipoprotein capsule containing microbubbles of a gas with a high molecular weight and low solubility in water. The researchers emphasized that these characteristics grant higher stability and longer time in circulation.

One of the main advantages of CEUS is the possibility of real time studies when compared to other contrast-enhanced exams such as magnetic resonance imaging (MRI) and computed tomography (CT), where evaluation occurs after the injection of the contrast medium [47] and the evaluation of nonsedated dogs when compared to CT and MRI [46].

CEUS is a quali-quantitative evaluation based on the study of the enhancement of patterns and the measurement of perfusion parameters, such as mean transit time (time from contrast injection until its complete clearance) [49].

A time–intensity curve is used to study some other functional parameters related to tissue perfusion. It evaluates the peak intensity (maximum enhancement), time to peak enhancement after contrast injection, time to peak after the contrast level is above baseline, upslope and downslope (regression lines of the time-intensity curve between 10% above the baseline to 85% of peak intensity). Upslope and downslope indirectly reflect the wash-in (contrast uptake) and wash-out (contrast clearance) rates based on estimates of 85% of the measured peak [51].

## 4. Applicability of CEUS

### 4.1. Male Reproductive Tract

There are several studies describing the use of contrast-enhanced ultrasound to evaluate the male reproductive tract in dogs. A study described the ultrasonographic aspect of different testicular tumors with CEUS [45]. These authors observed that testes with interstitial cell tumors presented inhomogeneous enhancement patterns with focal hyperechoic lesions in many of the cases. The same study reported that testes with seminoma presented homogeneous hyperechoic sign, persistent intra-tumoral vessels and iso- or hypoechoic parenchyma. Other findings described Sertoli tumors as inhomogeneous, with focal hyperechoic homogeneous or heterogeneous lesions, and a hyperechoic peripheral rim. In general, this study stated that inhomogeneous testes with hyperechoic lesions were associated with malignancy, with 87% sensitivity and a 100% positive predictive value.

Another study evaluated testicular tumors in nonsedated dogs and reported that interstitial cell tumors were hyperechoic, homogeneous, or inhomogeneous, with a peripheral hyperechoic rim and evident intra-tumoral vessels [46].

A subtle difference in the vascularization patterns of different testicular tumors in dogs was reported; thus, the use of different imaging modalities was encouraged. CEUS was associated with color or power Doppler ultrasound, allowing the investigation of intra- or perilesional arteries [48].

Contrast-enhanced ultrasonography of the prostate gland was previously used to evaluate the vascular perfusion kinetics. The authors reported that malignant prostatic diseases presented higher percentage peak intensities and a longer time to peak when compared to normal dogs or to dogs with benign prostatic diseases [52].

Other authors found that the perfusion values for malignant prostatic diseases differed from those of normal dogs. Adenocarcinomas presented higher percentage peak intensities, with a mean of 23.7% and a mean time to peak of 26.9 s. The values for leiomyosarcoma were lower than those for normal dogs, with a percentage of peak intensity of 14.1% and time to peak of 41.3 s [53].

### 4.2. Mammary Glands

Physiological alterations in the perfusion, size and the ultrasonographic aspect of mammary glands during the estrous cycle were investigated in bitches and represent the basis for the detection of mammary gland pathologies [50]. According to this study, during diestrus, all mammary glands increased in thickness and were presented as heterogeneous (B-mode ultrasonography), with a heterogeneous enhancement pattern (CEUS). The authors pointed out that abdominal cranial mammary glands presented an increase in the average transit time between estrus and late diestrus and a decrease between the end of diestrus and anestrus. Another finding was that inguinal mammary glands presented higher times to peak during anestrus when compared to estrus.

A study compared advanced ultrasonography methods for the evaluation of mammary neoplasia, evaluating 300 nodules in dogs [23]. It stated that CEUS was able to analyze tumoral macro- and microcirculation (not correlated with malignancy) and found an 80% sensitivity and low specificity (16%) for malignancy detection.

CEUS enabled the differentiation and staging of mammary carcinoma in bitches [54]. The authors found that the wash-in time and time to peak were lower than 7.5 s and 13.5 s, respectively, indicated complex carcinoma (62% sensitivity and 60% specificity). Similarly, grade 1 mammary carcinomas showed low values for the perfusion time in this study. Another important consideration was that the increased perfusion time (wash in > 6.5 s, time to peak > 12.5 s, and wash out > 64.5 s) indicated grade 2 or 3 carcinomas, with 68% sensitivity and 62% specificity.

Mammary gland tumors can frequently metastasize to regional lymph nodes, as these lymph nodes act as filters for the dissemination of tumor cells [26]. Contrast-enhanced ultrasound may aid in the detection of metastatic lymph nodes and identification of sentinel lymph nodes, providing important information for prognosis and treatment.

### 4.3. Kidneys and Urinary Bladder

Kidneys present a three-phased contrast enhancement consisting of an early arterial phase—in which there is rapid contrast enhancement due to the arterial blood supply—followed by a cortical phase marked by intense and uniform enhancement of the renal parenchyma, and a medullary phase, when the pyramids gradually uptake contrast until they are isochoic with the cortex [51,55].

Different types of renal tumors presented specific characteristics when evaluated with CEUS [56]. These authors reported that renal carcinomas presented large tortuous arteries with early contrast enhancement when compared to a normal renal parenchyma. Comparatively, in the same study, histiocytic sarcomas and lymphomas were less vascularized, with smaller arteries and early wash out during the corticomedullary phase.

In the corticomedullary phase (late), renal carcinomas presented homo- or heterogeneous, iso- or hypoechoic enhancement patterns, with progressive wash out. Metastasis of hemangiosarcoma presented no contrast enhancement in any phase (neither arterial nor corticomedullary). In addition to some particularities of each tumor type, there were some overlapping findings among malignant and benign tumors [56].

In the urinary bladder, evaluated with CEUS, a faster transit time (wash in, peak, and wash out) was associated with malignant lesions when compared to that of inflammatory lesions [57].

Other authors pointed out that ill-defined tumoral margins and poor differentiation between the tumor and the adjacent healthy tissue, if in the presence of a vascularized urinary bladder wall, can be associated with infiltrative tumors. Additionally, homo- or heterogeneous hyperenhancement patterns were associated with tumors [58].

### 4.4. Lymph Nodes

Peripheral lymph nodes were evaluated with CEUS and power Doppler. CEUS was able to detect twice as many blood vessels as the power Doppler investigation. Lymphomatous nodes presented hilar vessel displacement, neovascularization, and loss of the hyperechoic rim. The majority of the lymph nodes presented moderate to good perfusion with a homogeneous perfusion pattern [59].

Another study encouraged the association of different imaging modalities for the detection of mandibular lymph node metastasis. CEUS was associated with strain elastography. Contrast-filling defects, detected by CEUS, and a high elasticity index (stiffness), obtained with strain elastography, were suggestive of nodal metastasis [20].

### 4.5. Spleen

A study investigated focal splenic lesions with CEUS and reported that a hypoechoic lesion during wash out, which is associated with tortuous vessels, was suggestive of malignancy (Figure 3). Meanwhile, benign lesions presented a perfusion pattern similar to the adjacent splenic parenchyma. The same study found that hemangiosarcoma was presented as a large mass, with no perfusion in any phase, surrounded by a hypervascular splenic parenchyma. Lymphosarcoma presented faster time to peak and early wash out, with a honeycomb enhancement pattern during wash out [60].

Comparatively, other authors attested that the differentiation of benign and malignant splenic lesions must be based on the vascular tortuosity instead of considering the echogenicity or persistent hypoperfusion. Additionally, hypoperfusion persistent during all contrast phases can suggest malignancy with 40% sensitivity, 80% specificity and 71% accuracy [61].

### 4.6. Gastrointestinal Tract

Gastric neoplasia in dogs can be characterized using B-mode ultrasonography as severe gastric wall thickening (exceeding 1.2 cm), marked loss of the wall layering, and involvement of adjacent structures (i.e., regional lymphadenomegaly and steatites) [62]. Other authors mentioned that focal gastric wall thickness with loss of the normal wall layering may suggest gastric neoplasms [63]. Those features (to a much lesser extent) can be observed in cases of inflammatory conditions, with the exception of the involvement of lymph nodes and steatites [62].

There may be some differences in the symmetry of the wall thickening or the echogenicity of the gastric wall among gastric adenocarcinoma and lymphoma but, in general, B-mode ultrasonography findings may be nonspecific to distinguish different gastric tumors [64]. For that reason, contrast-enhanced ultrasonography is a promising modality that can help fill this gap, providing additional information to differentiate gastric neoplasms.

Malignant gastric tumors presented faster wash in compared to gastritis. B-mode ultrasound and CEUS were able to distinguish between malignant and benign gastric disorders, but the differentiation among several tumor histotypes still relies upon cytological or histopathological exams [62].

In the canine intestine, the most common tumors are carcinoma, lymphoma, leiomyoma and leiomyosarcoma [64]. There were no differences in the perfusion patterns between lymphoma and chronic inflammatory enteropathy or between lymphoma and the control group in dogs [65].

In felines, alimentary lymphoma, along with adenocarcinoma and mast cell tumors are the most common malignant neoplasia of the gastrointestinal tract [64]. CEUS and B-mode ultrasonography were used to differentiate lymphoma, gastritis, and normal stomachs in cats. There were overlapping findings between inflammation and low-grade lymphoma, both in the CEUS and B-mode ultrasound evaluations. High-grade lymphoma presented well defined characteristics such as thicker gastric walls with poor layer definition, a marked contrast-enhancement pattern, regional lymphadenopathy, and local steatitis [66].

### 4.7. Liver and Biliary System

In the canine liver, a study of hepatocellular carcinoma evaluated with CEUS described some variation in the tumoral presentation (i.e., contrast enhancement) according to the level of cellular differentiation. The size of the tumor had little influence on the pattern of enhancement, either during wash in or wash out [67].

There can be three different enhancement phases distinguished in the liver. The arterial phase starts after contrast injection and provides information on the arterial blood supply of a given lesion. The portal venous phase presents diffuse and maximal contrast enhancement of the hepatic parenchyma, and the late phase marks the decrease of the contrast enhancement until the complete clearance of the contrast from circulation [68].

Different types of hepatic tumors can present variable contrast-enhancement patterns. For that reason, cytology and histopathology are important to confirm the diagnosis. Most sarcomas presented no enhancement during wash in. Metastasis presented hyper-enhancement during wash in, with hypo-enhancement during wash out [69]. This indicates a lower portal supply or lower blood volume of these metastatic lesions when compared to the liver parenchyma [70]. A well-differentiated hepatocellular carcinoma was characterized with homogeneous hyper-enhancement during the arterial phase and homogeneous wash out [67].

There was an overlapping of qualitative findings with CEUS for different types of hepatobiliary neoplasia in cats. During the peak, biliary duct adenoma, biliary duct carcinoma, and hepatocellular carcinoma presented considerable variations in the echogenicity (hypo- or hyper-enhancement) in comparison to the normal hepatic parenchyma. Some biliary duct adenomas presented inhomogeneous hyper-enhancement during wash in (a characteristic of malignant tumors in dogs). There was no difference in the wash out between adenoma and biliary duct adenocarcinoma [69].

Adenomas, biliary duct carcinomas, and hepatocellular carcinoma shared similar characteristics during wash in and wash out in felines [71].

A hepatic mass in a cat, diagnosed as hemangiosarcoma (histopathology), was not identified by B-mode ultrasonography. However, CEUS was able to detect a hypoechoic mass during the contrast peak and portal phase [72].

Contrast-enhanced ultrasound enabled an increase in the differentiation of hepatic nodules and normal hepatic parenchyma. There was also a greater ability to detect malignant nodules when compared to B-mode ultrasonography. Benign nodules were less conspicuous, and there were no additional nodules detected after contrast enhancement [73]. An example can be seen in Figure 4.

Hypoechoic nodules detected in the hepatic parenchyma during the peak of contrast were highly suggestive of malignancy [73,74].

### 4.8. Adrenal Glands

Contrast-enhanced ultrasound studies evaluated adrenal gland neoplasia and established parameters for the differentiation of adenoma, adenocarcinoma, and pheochromocytoma [75,76,77,78]. Malignant adrenal gland tumors presented heterogeneous contrast-enhancement patterns. Carcinoma and pheochromocytoma presented lower regional blood volumes when compared to adenoma. Adenocarcinoma presented tortuous feeding vessels during the arterial and venous phases [75].

The mean transit time (for contrast) was significantly lower for malignant neoplasia than for adenomas [75]. Pheochromocytoma presented a faster time to peak and a shorter mean transit time than adenoma and adenocarcinoma and bigger upslope and downslope than adenocarcinoma [76].

The level of enhancement combined to the vascularization allowed the differentiation of malignant adrenal tumors (adenocarcinoma and pheochromocytoma) and benign adrenal tumors (adrenocortical adenoma) with 100% sensitivity, 80% specificity and 91.7% accuracy [77].

According to other authors, wash-out or perfusion patterns can be used to differentiate malignant or benign etiologies. Additionally, there was some overlap between those findings. Adenomas, adenocarcinomas, and pheochromocytoma shared similar characteristics, such as intra-lesional microcirculation and regions of hypoperfusion. In this way, cytology and histopathology are gold standard methods for confirming the diagnosis [78].

### 4.9. Pancreas

Contrast-enhanced ultrasound was used to investigate canine pancreatic neoplasia. This imaging modality allowed the differentiation of adenocarcinomas, insulinomas, and benign nodules. Adenocarcinomas presented hypoechoic contrast enhancement [79,80] and hypoperfusion [80], whereas insulinomas were presented as solid lesions, with a homogeneous and hyperechoic contrast enhancement [80] and uniform hyperperfusion [80].

Comparatively, nodular hyperplasia was isoattenuating to the surrounding pancreatic parenchyma, whereas cystic formations presented no contrast enhancement [80].

A study with three dogs described an increased conspicuity and better differentiation of pancreatic nodules (of insulinoma) after contrast injection. The enhancement pattern was very variable among the evaluated animals [81]. CEUS contributed to the detection of pancreatic nodules (of insulinoma) that were not detected by B-mode ultrasound [82].

Contrast-enhanced ultrasound of benign and malignant pancreatic nodules in cats showed that nodular hyperplasia was presented as small, hypoechoic nodules, isoechoic to the surrounding pancreatic parenchyma, with no wash-out phase. Cysts were anechoic, with thin layers and acoustic enhancement in B-mode ultrasonography. Those structures presented no enhancement with CEUS. Other benign lesions were similar to pseudocysts, with no intralesional vascularization. Pseudocyst-like lesions with intralesional vascularization were classified as adenocarcinomas. Adenocarcinomas and lymphomas presented large nodules with mixed echogenicity and hyper- or hypo-enhancement patterns [83]. The same study proved that contrast-enhanced ultrasound was sensitive and specific to differentiate nodular hyperplasia (100 and 94%), adenocarcinoma (85 and 77%) and other benign lesions (70 and 93%). However, this imaging modality was not able to differentiate lymphoma. The authors concluded that associating B-mode ultrasonography and CEUS can increase the accuracy to determine the etiology of the focal pancreatic lesions in cats. However, cytology and histopathology are paramount to confirm the diagnosis.

## 5. Conclusions

Elastography and contrast-enhanced ultrasonography provide important data on the differentiation of benign and malignant tumors in dogs and cats. These noninvasive imaging modalities are safe and can be easily performed on nonsedated animals, constituting interesting techniques for the investigation of neoplasia and metastasis in different tissues and organs. Their efficiency can be increased by the association with other imaging modalities such as B-mode or Doppler ultrasonography. Cytology and histopathology are the gold-standard methods to classify benign and malignant nodules and masses and determine the cellular type of a given neoplasm. However, they require invasive methods of tissue sampling, such as fine-needle aspiration or biopsy. In this way, elastography and CEUS, as well as other imaging modalities, can be used as screening tests, and can potentially represent an alternative to those invasive sampling methods.

## Figures and Tables

**Figure 1 animals-14-00312-f001:**
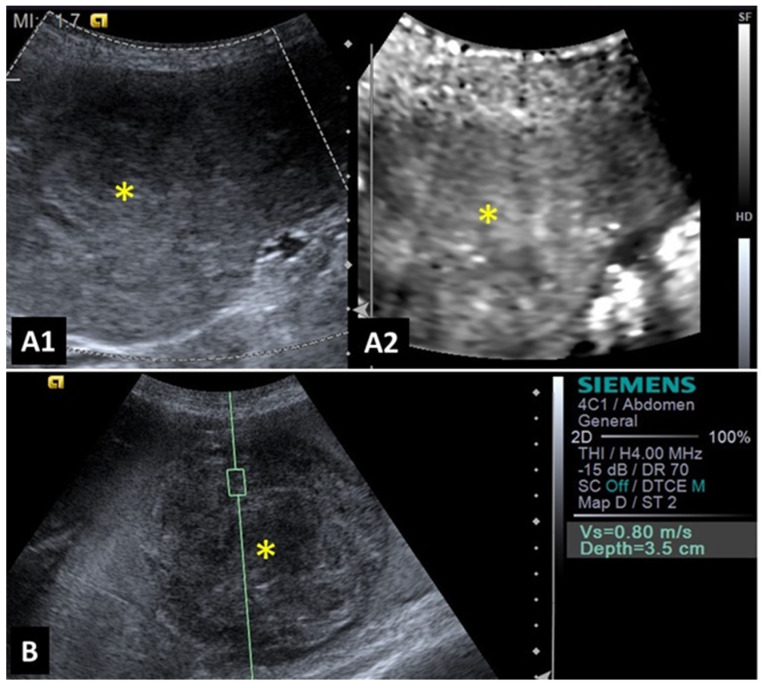
Image of a benign splenic lesion (hematoma) in a dog: (**A1**,**A2**) B-mode of the splenic lesion with mixed and heterogeneous echogenicity; (**B**) Acoustic Radiation Force Impulse (ARFI) elastography of the hematoma, demonstrating shades and shear velocity values indicative of decreased rigidity and benignity. Yellow asterisk indicates lesions.

**Figure 2 animals-14-00312-f002:**
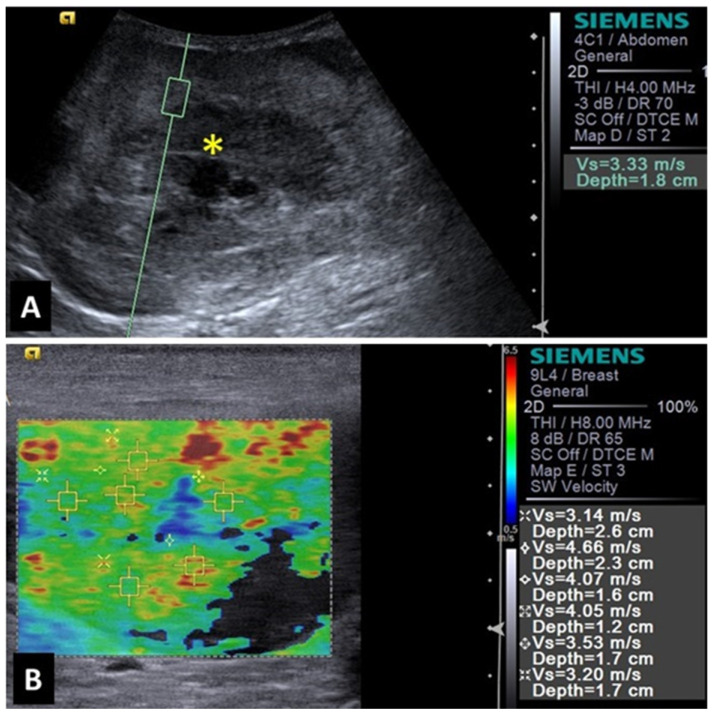
Image of malignant splenic lesion (hemangiosarcoma) in a dog: (**A**) B-mode of splenic lesion with mixed and heterogeneous echogenicity; (**B**) Acoustic Radiation Force Impulse (ARFI) elastography of hemangiosarcoma, demonstrating shades and shear velocity values indicative of increased rigidity and malignancy. Yellow asterisk indicates lesions.

**Figure 3 animals-14-00312-f003:**
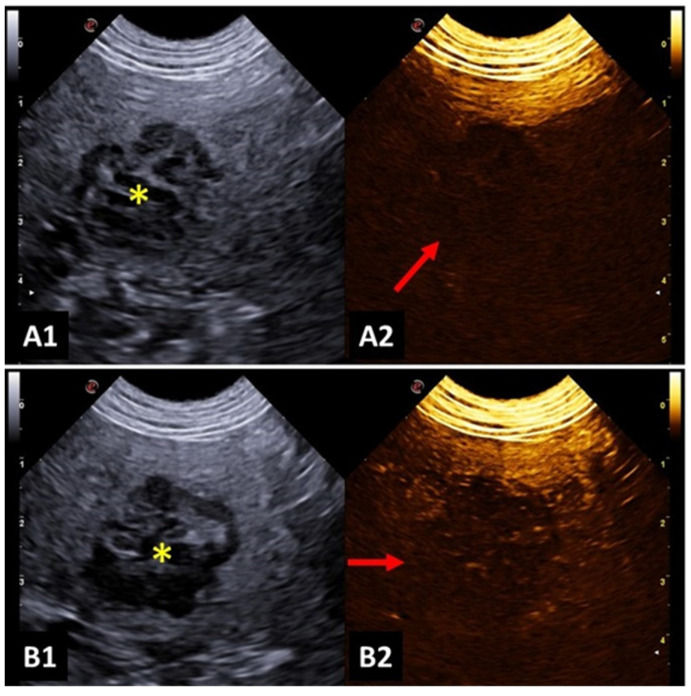
Image of malignant splenic lesion in a dog: (**A1**,**B1**) B-mode image of the splenic lesion with mixed and heterogeneous echogenicity; (**A2**) Contrast-enhanced ultrasonography (CEUS) image before contrast filling; (**B2**) hypointense splenic lesion during contrast wash out, indicating characteristics of malignancy. Yellow asterisk indicates lesions.

**Figure 4 animals-14-00312-f004:**
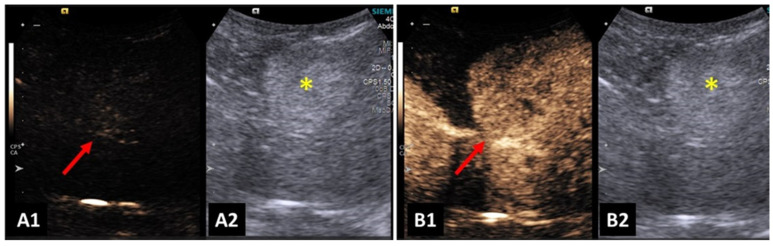
Image of a benign liver lesion in a dog: (**A1**) lesion before contrast filling—wash in; (**A2**,**B2**) B-mode image of the hyperechoic liver lesion; and (**B1**) homogeneous and hyperechoic liver lesion from Contrast-enhanced ultrasonography (CEUS), in phase of the peak enhancement. Yellow asterisk indicates lesions.

## Data Availability

The data presented in this study are available on request from the corresponding author for scientific purposes.

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
