# Peer review of "Use of New Ultrasonography Methods for Detecting Neoplasms in Dogs and Cats: A Review"

_animals, 2024, doi:10.3390/ani14020312_

Round 1

Reviewer 1 Report

Comments and Suggestions for Authors

Dear Authors

I reviewed the manuscript entitled "Use of new ultrasonography methods for detecting neoplasms 2 in dogs and cats: a review". The manuscript is well organized. However it results too concise in several paragraphs, and this can be confusing to the reader (see specific comments). Moreover, some references are missing. Therefore I recommend major revision.

Specific comments

Introduction to elastography

Please, add a more accurate description of strain and shear wave elastography and the differences between the two techniques. The section is confusing in my opinion.

Line 97: please be consistent with the terms. ARFI elastosonography, shear wave elastosonography, sonoelastography...these terms could be confusing to the reader.

lines 146-153: see comment above. Terms related to elastography and the description of the lymph nodes pattern mixed with CEUS pattern are confusing.

Lines 276-278: please briefly describe the perfusion parameters, adding mean transit time, upslope and downslope and describe be perfusion curve

pararagraph 4.1

line 285-290: please change hyperintense into hyperenhanced or hyperechoic

line 299-300: There are several reports describing the use of contrast-enhanced ultrasound in prostatic tumors in dogs : 

-B-mode and contrast-enhanced ultrasonographic findings in canine prostatic disorders

M Russo , M Vignoli, G C W England 

Reprod Domest Anim 2012 Dec:47 Suppl 6:238-42.

- Assessment of vascular perfusion kinetics using contrast-enhanced ultrasound for the diagnosis of prostatic disease in dogs.

Vignoli M, Russo M, Catone G, Rossi F, Attanasi G, Terragni R, Saunders J, England G.

Reprod Domest Anim. 2011 Apr;46(2):209-13.

lines 323-326: please briefly describe the phases of renal enhancement.

Paragraph 4.7: please add a description of liver perfusion. Hypoenhancement in portal and late phase is a sign of malignancy, this should be stressed in this paragraph

Paragraph 4.8 Adrenal glands

line 423: "Carcinoma and pheochromocytoma presented lower retinal blood volume when compared to adenoma". This sentence is obscure to me. What does "retinal blood volume" mean?

line 428: what do the Authors mean with "bigger upslope and downslope"?

Figure 4: please add a more thorough description of the phases of this liver lesion enhancement 

Author Response

Author's Reply to the Review Report (Reviewer 1)

Comments and Suggestions for Authors

Dear Authors

I reviewed the manuscript entitled "Use of new ultrasonography methods for detecting neoplasms 2 in dogs and cats: a review". The manuscript is well organized. However it results too concise in several paragraphs, and this can be confusing to the reader (see specific comments). Moreover, some references are missing. Therefore I recommend major revision.

Dear Reviewer 1,

We would like to thank you for the attention with our manuscript and for the important considerations. We believe that those modifications will improve clarity and quality of our manuscript. Please find our specific answers to each of your comments below. Please note that the line numbers have changed due to the insertion of new text fragments. We have added the line numbers for each modification along with each answer.

Specific comments

Introduction to elastography

Please, add a more accurate description of strain and shear wave elastography and the differences between the two techniques. The section is confusing in my opinion.

  1. Modifications made in the section "Introduction to elastography", lines 55-68.

Line 97: please be consistent with the terms. ARFI elastosonography, shear wave elastosonography, sonoelastography...these terms could be confusing to the reader.

  1. We have made the adjustments accordingly in the section "Introduction to elastography", lines 55-84.

lines 146-153: see comment above. Terms related to elastography and the description of the lymph nodes pattern mixed with CEUS pattern are confusing.

  1. We have made modifications according to your suggestions in lines 162-173

Lines 276-278: please briefly describe the perfusion parameters, adding mean transit time, upslope and downslope and describe be perfusion curve

  1. Descriptions inserted in lines 293-302

Paragraph 4.1

line 285-290: please change hyperintense into hyperenhanced or hyperechoic

  1. Changes added to the lines 309-315

line 299-300: There are several reports describing the use of contrast-enhanced ultrasound in prostatic tumors in dogs:

-B-mode and contrast-enhanced ultrasonographic findings in canine prostatic disorders

M Russo , M Vignoli, G C W England Reprod Domest Anim 2012 Dec:47 Suppl 6:238-42.

  • Assessment of vascular perfusion kinetics using contrast-enhanced ultrasound for the diagnosis of prostatic disease in dogs.

Vignoli M, Russo M, Catone G, Rossi F, Attanasi G, Terragni R, Saunders J, England G. Reprod Domest Anim. 2011 Apr;46(2):209-13.

  1. We have added these additional citations according to your suggestions. Please check lines 323-331

lines 323-326: please briefly describe the phases of renal enhancement.

  1. Description added, please check lines 359-363

Paragraph 4.7: please add a description of liver perfusion. Hypoenhancement in portal and late phase is a sign of malignancy, this should be stressed in this paragraph

  1. Description added to the text, please check lines 444-449. Paragraph detailed in lines 454-457

Paragraph 4.8 Adrenal glands

line 423: "Carcinoma and pheochromocytoma presented lower retinal blood volume when compared to adenoma". This sentence is obscure to me. What does "retinal blood volume" mean?

  1. We apologize for this typographical error. We meant "regional blood volume" , lines 481-482

line 428: what do the Authors mean with "bigger upslope and downslope"?

  1. Upslope and downslope are indirectly related to wash-in and wash-out based on estimates of 85% of the intensity peak. The new sentence in lines 486-487 should be interpreted as:

"Pheochromocytoma presents bigger wash-in and wash-out rate* compared to adenocarcinoma."

Authors decided not to describe upslope and downslope here considering that a proper description was inserted in the section "Introduction to CEUS". We would be happy to make any further adjustment or modification if needed.

Figure 4: please add a more thorough description of the phases of this liver lesion enhancement

  1. added

Reviewer 2 Report

Comments and Suggestions for Authors

Dear authors,

Congratulations, overall it's a work well done. What I have included are papers that in my opinion are necessary to cite and discuss. The content of the review can actually be expanded and I am convinced that with a little work you can significantly improve the quality of the paper.

Author Response

Author's Reply to the Review Report (Reviewer 2)

Comments and Suggestions for Authors

Dear authors,

Congratulations, overall it's a work well done. What I have included are papers that in my opinion are necessary to cite and discuss. The content of the review can actually be expanded and I am convinced that with a little work you can significantly improve the quality of the paper.

Dear Reviewer 2,

We would like to thank you for the attention with our manuscript and for the important considerations. We believe that those new citations will greatly improve manuscript quality. Please find our specific answers to each of your comments below. Please note that the line numbers have changed due to the insertion of new text fragments. We have added the line numbers for each modification along with each answer.

Line 85: Is it possible to insert a reference for this statement? I noticed several references already included in the text that could be appropriate, but I think it is an important introductory sentence that must be supported by a bibliographical source. I would recommend Doi: 10.3390/ani11041115. PMID: 33924625; PMCID: PMC8070006;

  1. Reference inserted to the lines 96-97

Line 99-102: English check;

  1. We have made the requested modifications - lines 110-114

Line 115- 165 Please consider considering this literature as an interesting starting point to include on the lymph node section: Doi: 10.3390/vetsci9090484; 10.3390/ani11030883; PMID: 36733656; PMCID: PMC9847431; 10.1016/j.cvsm.2019.04.003; DOI:10.1111/vsu.13537; DOI:10.1111/vsu.13929; DOI:10.1007/s11864-022-00999-5; DOI:10.1111/vco.12592;

  1. We have added the new citations to the lines 126-131

Line 367: For gastroenteric tract I think is important to cite and discuss this review 10.3844/ajavsp.2020.89.101;

  1. We have added this suggested review to the section "4.6 Gastrointestinal tract"

Line 301: Please consider including this important reference. Insert it and comment on it doi 10.3390/ani11041115;

  1. Citation added to the lines 354-357

Round 2

Reviewer 1 Report

Comments and Suggestions for Authors

Dear Authors,

The manuscript is improved and my concerns have been addressed accordingly.

Please check line 65 for the reference "ref: cap que o professor nos mandou"